# Nuclease Triggered “Signal-On” and Amplified Fluorescent Sensing of Fumonisin B_1_ Incorporating Graphene Oxide and Specific Aptamer

**DOI:** 10.3390/ijms23169024

**Published:** 2022-08-12

**Authors:** Xiaodong Guo, Qinqin Qiao, Mengke Zhang, Marie-Laure Fauconnier

**Affiliations:** 1School of Agriculture and Biology, Shanghai Jiao Tong University, Shanghai 200240, China; 2Chimie Générale et Organique, Gembloux Agro-Bio Tech, Université de Liège, Passage des Déportés 2, 5030 Gembloux, Belgium; 3Laboratory of Quality and Safety Risk Assessment for Dairy Products of Ministry of Agriculture and Rural Affairs, Institute of Animal Science, Chinese Academy of Agricultural Sciences, Beijing 100193, China; 4College of Information Engineering, Fuyang Normal University, Fuyang 236041, China

**Keywords:** aptasensor, fumonisin B_1_, nuclease, graphene oxide, point-of-care testing, food safety

## Abstract

Remarkable advancements have been achieved in the development of rapid analytic techniques toward fumonisin B_1_ (FB_1_) monitoring and even trace levels for food safety in recent years. However, the point-of-care testing for quantitative and accurate FB_1_ determination is still challenging. Herein, an innovative aptasensor was established to monitor FB_1_ by utilizing graphene oxide (GO) and nuclease-triggered signal enhancement. GO can be utilized as a fluorescence quenching agent toward a fluorophore-modified aptamer, and even as a protectant of the aptamer from nuclease cleavage for subsequent target cycling and signal amplification detection. This proposed sensing strategy exhibited a good linearity for FB_1_ determination in the dynamic range from 0.5 to 20 ng mL^−1^ with a good correlation of R^2^ = 0.995. Its limit of detection was established at 0.15 ng mL^−1^ (S/N = 3), which was significantly lower than the legal requirements by three orders of magnitude. The interferent study demonstrated that the introduced aptasensor possessed high selectivity for FB_1_. Moreover, the aptasensor was successfully applied to the detection of wheat flour samples, and the results were consistent with the classical ELISA method. The rapid response, sensitive and selective analysis, and reliable results of this sensing platform offer a promising opportunity for food mycotoxin control in point-of-care testing.

## 1. Introduction

Mycotoxin contamination in food is of worldwide concern, and poses serious hazards to human health [1,2,3,4]. Fumonisins, an important molecule group of carcinogenic mycotoxins, mainly occur through fungal species such as *Fusarium moniliforme* and *Fusarium proliferatum* composed of various tricarballylic acid and polyhydric alcohol [5]. Of the major fumonisins, fumonisin B_1_ (FB_1_) is the most toxic and present one, accounting for 70% of total fumonisin contamination [6,7,8]. Consequently, the International Agency for Research on Cancer (IARC) has categorized FB_1_ as a 2B group carcinogen [9,10]. Accordingly, the United States Food and Drug Administration (FDA) has regulated the maximum residue limit (MRL) for whole fumonisins (sum of FB_1_, FB_2_, and FB_3_) as 2 mg kg^−1^ in degermed dry-milled corn products [11], and the MRL value for combined FB_1_ and FB_2_ set by European Union was restricted to 1 mg kg^−1^ in maize [12]. Considering the low MRL and the enhancement of toxic damage, rapid, accurate, sensitive, and selective analytical techniques of FB_1_ detection are urgently required to ensure food safety.

For the monitoring of trace levels of FB_1_, analytical approaches are mainly based on high-performance liquid chromatography (HPLC) [13,14], high-performance liquid chromatography combined with mass spectrometry (HPLC–MS) [15,16,17], and classic immunoassays [18,19,20,21]. However, these techniques commonly suffer from some limitations such as high cost, highly trained personnel, low stability, as well as complicated protocols. To overcome the barriers, great endeavors have been performed to develop a fluorescent methodology for food safety. Moreover, aptamers, owing to their distinguishing characteristics such as ease of modification and high specificity, etc., have been confirmed to be similar or even superior to antibodies [22,23,24]. Aptamer-based fluorescent sensing has been established towards FB_1_ [25,26]. Nevertheless, these sensing strategies commonly require the conjugation of the aptamer with probes, as well as complicated protocols. Accordingly, the development of point-of-care (POC) sensing platforms for rapid and sensitive FB_1_ analysis remains challenging.

Graphene oxide (GO) has been a rising star nanomaterial for sensing applications in recent years [27,28,29]. Excitingly, single-stranded DNA (ssDNA) aptamers can be directly modified with fluorophores to produce a fluorescent signal, which would be quenched by GO via π–π stacking interactions between fluorophores and GO [28,30,31]. In addition to the fluorescence quenching performance, GO can protect ssDNA aptamers from nuclease digestion because of the hydrophobic stacking reactions between nucleobases and GO [32,33,34]. As a consequence, fluorescent aptasensing coupled with GO nanomaterials has been developed to monitor AFM1 and AFB1 in our previous research and another recent attempt, respectively [35,36]. To the best of our knowledge, an aptamer-based sensor combining fluorescence-quenching and aptamer protection of GO with nuclease amplification for detection of FB_1_ has not yet been found.

Inspired by this knowledge, a novel nuclease triggered “signal-on” and amplified fluorescent sensing of FB_1_ was fabricated using GO nanomaterial and a specific aptamer. The embedding of GO was realized for fluorescence quenching and the protectant of aptamers from nuclease cleavage. In the absence of FB_1_, the introduction of GO can avoid the digestion of aptamers by nuclease, and the “signal-off” mechanism was induced. When target FB_1_ was present, the aptamer could capture the target to form a special three-dimensional configuration, resulting in the separation of the aptamer from the GO surface. Then, the aptamer was digested by nuclease and released FB_1_, and target cycling signal amplification was eventually achieved. Consequently, the quantitative detection of FB_1_ levels was established via monitoring the changes in fluorescent signals within 5 min.

## 2. Results and Discussion

### 2.1. Sensing Strategy for FB_1_ Detection

As mentioned in Section 1, graphene oxide binds to ssDNA such as aptamers with high efficiency as a result of π–π stacking and hydrophobic interaction. As a consequence, the fluorescence signal of the fluorophore-modified aptamer was dramatically reduced owing to GO’s powerful fluorescence quenching property. GO can be thus integrated in aptasensing construction on food hazards detection. Moreover, to enhance the signal response, the nuclease (DNase I) was embedded to digest the aptamer into DNA fragments, leading to the release of FB_1_. A schematic representation of this aptasensor for amplified FB_1_ detection was depicted in Figure 1. In this novel design, the specific aptamer was modified with fluorophore carboxy-X-rhodamine (ROX). Upon the addition of aptamer into GO solution, the fluorescence signal was significantly decreased, which revealed great adsorption and fluorescence quenching of GO toward the aptamer. When FB_1_ was present, the aptamer preferred to bind the target, generating a special three-dimensional configuration. Subsequently, the aptamer was separated and digested by the nuclease. The target was then released from the compound and available for recognition by another sequence. Hence, a cycling signal amplification was realized for the highly sensitive detection of FB_1_.

### 2.2. Signal Enhancement Sensing of FB_1_ with Nuclease

As shown in Figure 2, when GO was present at 20 μg mL^−1^, the fluorescent intensity was dramatically reduced. Once the FB_1_ level reached 10 ng mL^−1^, the fluorescent signal was increased, which demonstrated the generation of an aptamer/FB_1_ compound, and the separation of the aptamer. In addition, the molecule recognition of the aptamer was not affected by fluorophore modification. Upon the simultaneous addition of FB_1_ and nuclease, a significant enhancement of the fluorescent signal by 110% over the background was measured, indicating that the embedding of nuclease led to the enhancement of the fluorescent signal, together with the improvement in the signal-to-noise (S/N) ratio and an in the amplified detection of FB_1_.

### 2.3. Detection Performance of the Aptasensor

The analytical performance of the proposed amplified aptasensing platform was evaluated by the analysis of the fluorescence signal response versus different levels of FB_1_. The detection conditions were 585 nm of the excitation wavelength and 605 nm of the emission wavelength. As illustrated in Figure 3, it can be seen that the fluorescent intensity was enhanced as the increase in target concentrations in the range of 0.5–20 ng mL^−1^. Moreover, a dynamic response was observed between the fluorescent signal and target levels. The linear equation was achieved as F = 31.65 C + 126.05 with a high correlation of R^2^ = 0.995, where F represents the fluorescence signal intensity and C represents concentrations of FB_1_. The limit of detection (LOD) was calculated to be 0.15 ng mL^−1^ (signal-to-noise = 3), demonstrating a wide linear response and compatible detection sensitivity toward FB_1_ in comparison with the protocols reported previously (Table 1). In particular, the proposed method exhibited relatively low LOD over the antibody-based immunosensors and other fluorescent aptasensors [9,20,25,37,38,39,40,41,42]. Additionally, it is well known that the production and preparation of antibodies has a high cost and a long period. Antibody-based immunoassays are pretty expensive. However, the synthesis and modification of aptamer (25–80 bases) can be completed by the biotech company with only a few dollars. Especially, the cost of the fluorescent aptasensor in another attempt is also more expensive than that of this work since noble metal platinum nanoparticles (Pt NPs) are required in their design [25]. Therefore, the proposed method is cheaper than the existing ones. More excitingly, only 5 min is required in the analytic process, demonstrating that the promising point-of-care testing of mycotoxins is superior to other analytic techniques.

### 2.4. Selectivity Analysis of the Aptasensor

Selectivity validation plays a very important role in the preciseness assessment of the developed aptasensor. To assess the selectivity of the developed aptasensor for FB_1_ determination, other mycotoxins such as AFB_1_, AFM_1_, and OTA were measured in this sensing protocol with the same level (5 ng mL^−1^) as that of FB_1_. In addition, the detection procedures were also under identical experimental conditions as FB_1_ detection. As seen in Figure 4, the proposed aptasensor displayed a strong fluorescent signal to monitor FB_1_. When other mycotoxins were added, the fluorescent signal was significantly reduced, and a similar result was obtained in the control group. These results confirm the specificity of the aptamer for the recognition of FB_1_. Furthermore, the results obtained in this section reveal that this sensing platform possesses satisfactory specificity for FB_1_ analysis.

### 2.5. Method Validation of This Method

The applicability of the sensing strategy was investigated for the detection of FB_1_ in wheat flour samples. The results in Table 2 showed that the recovery ratios in the range of 99% to 111% were monitored in the spiked wheat flour samples, which were satisfactory for mycotoxin monitoring by using a rapid screening method. Meanwhile, the detection results measured by the classic ELISA method ranged from 100% to 114%, demonstrating the high agreement with the current aptasensing strategy for detecting similar samples. It was further revealed that this method was accurate and reliable for FB_1_ analysis in real samples, and moreover, provided a promising potential in hazards detection to ensure food safety.

## 3. Materials and Methods

### 3.1. Materials and Reagents

Ochratoxin A (OTA), Aflatoxin B_1_ (AFB_1_), aflatoxin M_1_ (AFM_1_), and FB_1_ standard substances were obtained from Sigma-Aldrich (St. Louis, MI, USA). In addition, graphene oxide and DNase I (RNase-free) were purchased from Sigma-Aldrich (St. Louis, MI, USA). Chemicals materials, namely, sodium chloride (NaCl), potassium chloride (KCl), anhydrous calcium chloride (CaCl_2_), and 2-Amino-2-(hydroxymethyl)-1,3-propanediol (Tris) were from Shanghai Chemical Reagent Company (Shanghai, China). All chemicals used in this experiment at least were analytical grade and used as received with no further purification. Double-distilled water was used throughout the study. The specific aptamer oligonucleotides synthesized by Sangon Biotech., Co., Ltd. (Shanghai, China) were purified through the HPLC system and utilized in the experiment. The aptamer stock solutions were obtained using Tris buffer (10 mM Tris, 120 mM NaCl, 5 mM KCl, 20 mM CaCl_2_, pH 7.0). As shown in Figure 5, the ssDNA aptamer oligonucleotides of the carboxy-X-rhodamine (ROX)-modified FB_1_ aptamer and the specific interactions with the target are illustrated [43].

### 3.2. Fluorescent Response for Aptasensing of FB_1_

To achieve amplified monitoring of FB_1_ [35,36], the fluorophore-modified aptamer was dissolved and diluted to 100 nM with Tris buffer. Then, graphene oxide at a concentration of 20 μg mL^−1^ was incubated with the aptamer solution at room temperature for 15 min to produce an aptamer/GO compound (Figure 5), together with a remarkably reduced fluorescent signal. Subsequently, various levels of target FB_1_ and DNase I (100 U) were added to the mixture simultaneously. Next, the complex was incubated for signal enhancement at room temperature for 1 h. Ultimately, the Shimadzu RF-5301 Luminescence Spectrophotometer (Tokyo, Japan) was used to record the fluorescent intensity. The experiment conditions were under the excitation wavelength of 585 nm, and the emission spectra were measured in the wavelength range of 590–690 nm. Slit widths for both the excitation and emission were set at 10 nm.

### 3.3. Specificity Analysis

To investigate the performance of this aptasensing method for the highly selective recognition of FB_1_ over other substances, mycotoxin standard substances including AFB_1_, AFM_1_, and OTA were respectively measured at the same concentration of 5 ng mL^−1^. The analytical protocol was identical to that of FB_1_ determination.

### 3.4. Practicability Analysis of This Aptasensing Platform

The proposed aptasensing method was realized for quantitative detection of FB_1_ in wheat flour samples for practicability analysis. The prepared samples were spiked with 2 mL of FB_1_ at concentrations of 0, 1.5, 8, and 15 ng mL^−1^, respectively, and were operated in triplicate, achieving final levels of 0, 1.5, 8, and 15 μg kg^−1^. Each sample was accurately weighed (2.00 ± 0.05 g), and extraction of the samples was performed with 2 mL of extraction solution (50% methanol in water). Subsequently, the obtained mixtures were filtrated via a syringe filter (0.45 μm) three times. Eventually, the filtrates were collected and monitored by the amplified aptasensing experiments and the ELISA method.

### 3.5. Statistical Analysis of the Experiment Results

Standard deviations (SDs) and means of fluorescent intensities were achieved in triplicate. The calibration curve standards and samples for detection of FB_1_ were performed from three replicates. Fluorescence emission spectra curves toward FB_1_ determination were plotted by using Origin 8.0 software (OriginLab Corporation, Northampton, MA, USA). Linear regression analysis was achieved with Microsoft Excel between fluorescent signals and concentrations of FB_1_.

## 4. Conclusions

In this work, a novel, sensitive, and accurate aptasensor for amplified and specific detection of FB1 was firstly introduced, which relies on the GO and DNase I-induced target cycling and signal enhancement strategies. A wide dynamic range from 0.5 to 20 ng mL^−1^ was achieved between the fluorescence intensity and concentrations of FB1, its detection of limit was determined to be 0.15 ng mL^−1^, which is sensitive and compatible with the current methods. In addition, the specific tests and practical analysis performance were also investigated by detecting different mycotoxins and real wheat flour samples. Compared to the previous methods reported in the literature, this novel fluorescent sensing platform exhibited advantages such as ease of operation, excellent sensitivity, and selectivity, as well as low cost (several hundred dollars). Moreover, this proposed approach allowed point-of-care testing since it only took 5 min to complete the analysis detection; in particular, it is well-known that hand-held fluorometers, cover the emission spectra in the range 590–690 nm, and have been widely developed in fluorescent sensing platform. Therefore, the fabricated aptasensor coupled with hand-held fluorometers opens up a new horizon for on-site detection of FB1. Given the promising potential of this developed fluorescent aptasensor, future studies are expected to improve the detection efficiency and applicability for food safety.

## Figures and Tables

**Figure 1 ijms-23-09024-f001:**
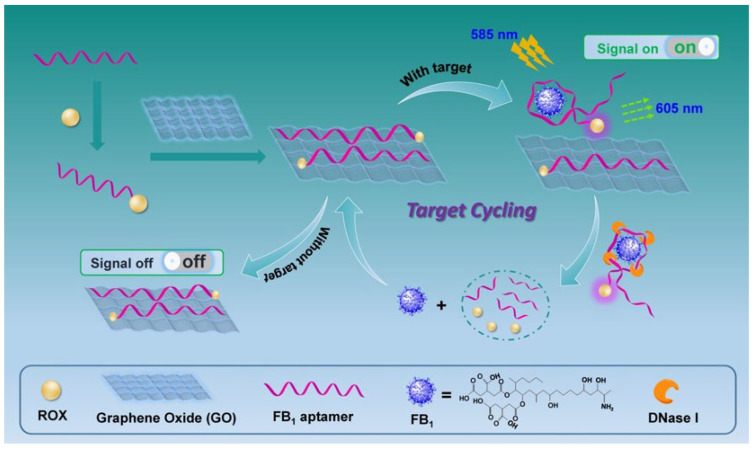
Schematic representation of the GO-assisted fluorescent aptasensor platform for detection of fumonisin B_1_ via the utilization of nuclease triggered signal-on performance and the specific aptamer.

**Figure 2 ijms-23-09024-f002:**
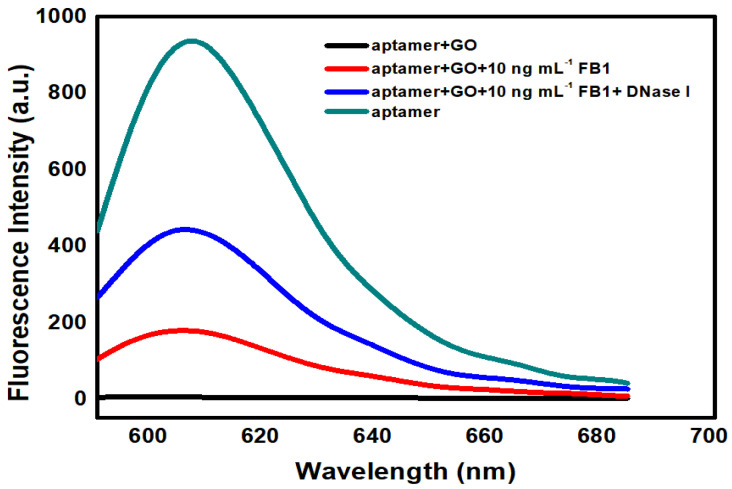
Fluorescence emission spectra of this sensing method in different conditions including the absence (0) of FB_1_, presence of 10 ng mL^−1^ FB_1_, and 10 ng mL^−1^ FB_1_ and 100 U DNase I. Conditions: 100 nM FB_1_ aptamer, 20 μg mL^−1^ GO in Tris buffer (10 mM Tris, 120 mM NaCl, 5 mM KCl, 20 mM CaCl_2_, pH 7.0). Excitation wavelength (λex) is set at 585 nm.

**Figure 3 ijms-23-09024-f003:**
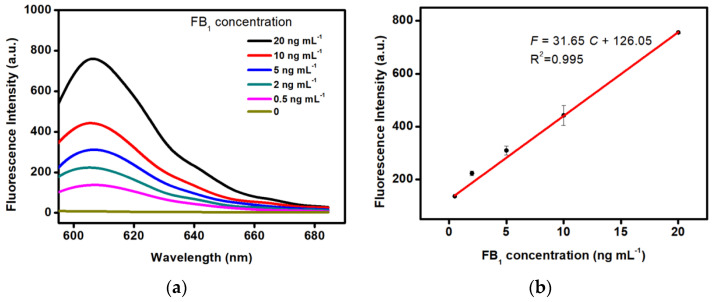
(**a**) Fluorescence emission spectra of the aptasensor in the addition of FB_1_ at various concentrations. (**b**) Linear relationship between the fluorescence intensity and FB_1_ concentrations in the range of 0.5 to 20 ng mL^−1^.

**Figure 4 ijms-23-09024-f004:**
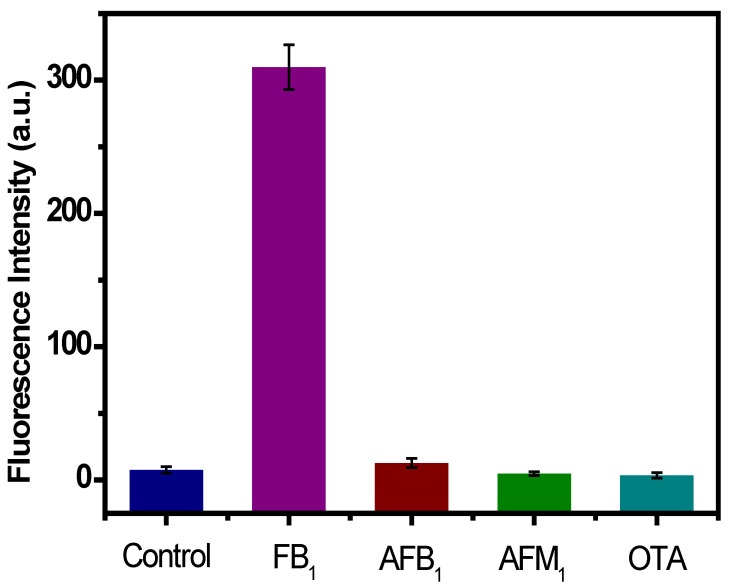
Fluorescence signal response in the absence (control) and presence of mycotoxins at a concentration of 5 ng mL^−1^: FB_1_, AFM_1_, AFB_1_, and OTA. The measurement conditions were as follows: Excitation wavelength (λex) was set at 585 nm, 100 nM FB_1_ aptamer, 20 μg mL^−1^ GO, 100 U DNase I. Each data point was the mean of three replicates.

**Figure 5 ijms-23-09024-f005:**
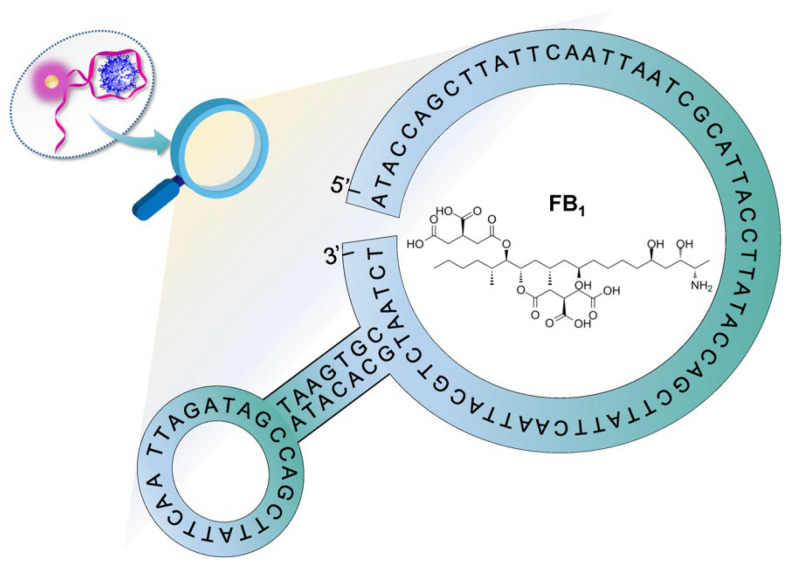
Illustration of the aptamer and its specific interactions with target fumonisin B_1_.

**Table 1 ijms-23-09024-t001:** Comparison of the analytical performance of currently available methods for the detection of FB_1_.

Method	Detection Time (min)	Linear Range (ng mL^−1^)	LOD (ng mL^−1^)	Reference
Chemiluminescence ELISA	60	0.93–7.73	0.12	[37]
Electrochemical	180	0.01–1000	0.002	[20]
Amperometric	60	0.73–11.2	0.33	[38]
ELISA	~60	0.27–5.92	0.15	[39]
Chemiluminescence	60	0.01–0.1	0.0017	[40]
Chemiluminescence	150	0.05–25	0.027	[41]
Colorimetric immunoassay	120	3.125–25	12.5	[9]
Antibody-based HRP sensor	22	0.31–162.42	0.21	[42]
Fluorescent aptasensor	15	1–10,000	0.4	[25]
Fluorescent aptasensor	5	0.5–20	0.15	Current work

**Table 2 ijms-23-09024-t002:** Detection of FB_1_ in the wheat flour samples.

Sample	Spiked Concentration (ng mL^−1^)	Current Aptasensor Method		Classic ELISA Method	
		Detected Concentrations Mean ^a^ ± SD ^b^ (ng mL^−1^)	Recovery (%)	Detected Concentrations Mean ^a^ ± SD ^b^ (ng mL^−1^)	Recovery (%)
Wheat flour	0	ND ^c^	-	ND ^c^	-
	1.5	1.67 ± 0.02	111	1.71 ± 0.08	114
	8	7.93 ± 0.56	99	8.02 ± 0.52	100
	15	15.47 ± 0.68	103	16.22 ± 0.84	108

^a^ The mean of three measurements; ^b^ SD means standard deviation; ^c^ ND means not detected.

## Data Availability

The data that support the findings of this study are available from the corresponding author on reasonable request.

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
