# Peer review of "Nuclease Triggered “Signal-On” and Amplified Fluorescent Sensing of Fumonisin B1 Incorporating Graphene Oxide and Specific Aptamer"

_ijms, 2022, doi:10.3390/ijms23169024_

Round 1

Reviewer 1 Report

The authors report on a new method based on graphene oxide and nuclease triggered signal enhancement. Generally speaking, the topic of graphene based aptasensors has increased interest; due to the high importance and demand on sensors for pollutants and contaminations for all kind of fields, and the level of the current state-of-the-art regarding   scientific and application related aspects is high. The authors proposed a cheaper analytical approach for the detection of fumonisin B1 using nuclease amplifications.

Scheme 1 explain their theory and the manuscript is well written and structured, almost each claim being supported by results.

The authors compare the analytical performance of their method with currently available methods in table 1, showing that their results are competitive.

Couple small modification should be added to the manuscript.

The authors mention that the proposed method is cheaper than the existing ones, but except mentioning that it costs several hundred dollars, they don’t compare prices with the other methods.

A claim not quite correct is that the method will only require 5 minutes, but an hour is needed for the complex incubation beforehand. 

To check the selectivity of the developed aptasensor for FB1  determination, other mycotoxins - AFB1, AFM1, and OTA were employed. Can the authors explain why they choose these mycotoxins? What will be the aptasensor response for FB2 and FB3 mycotoxins, that have similar structures?

Table 2 shows the detection of FB1 in wheat flour samples. The results from ELISA method should also be added

Reviewer 2 Report

The paper is aimed on the development of analytical methods of determination of Mycotoxin contamination in food.  It is worldwide known that fumonisin B1 (FB1) possess serious hazards to human health.  In the proposed paper an innovative aptamer-based fluorescent sensing has been established towards FB1 by using graphene oxide (GO) and nuclease triggered signal enhancement.

This study firstly introduced an innovative aptasensor for amplified fluorescent detection of FB1, which involved aptamer as the molecule recognition and GO as a fluorescent quencher with fluorescent recovery mechanism. This signal-on approach achieved by nuclease-induced target cycling signal enhancement allowed a wide dynamic response  in the range of 0.5-20 ng mL-1 toward FB1, and its limit of detection was achieved at 0.15 ng mL-1 . Moreover, this developed aptasensor exhibited excellent selectivity for FB1 over other mycotoxinsThe rapid response, sensitive and selective analysis, and reliable results of this sensing platform offer a promising opportunity for food mycotoxins control in point-of-care testing.

 The paper describes new innovative results aimed on practical applications and can be published in resent form after small English corrections.

Reviewer 3 Report

The authors report the development of a fluorescent aptamer-based sensing platform for the detection of fumonisin B1 (FB1) mycotoxin. The work is interesting and the basis of the research is clear; anyway, there are several aspects of the work that need to be addressed.

First of all, I would suggest to extend the introduction, explaining more in deep the background of the research and giving more importance to the cited literature. The presented assay has also to be discussed more in deep, underlining the importance of the reported strategy in, for example, food and environmental analysis, which is cited in the abstract but not fully addressed in the body text. Just some spare sentences are not enough to build a good paper.

What is really missing, though, is the whole optimization of the experimental procedure. Why did the authors choose to use, for instance, exactly 20 mg/L of GO or to incubate the aptamer/FB1 complex for 1 h? Why were FB1 and nuclease added together to the mixture? and so on. Please extend this part.

As the scheme of the presented assay is quite simple, the authors should add some novelty/characterization to their work, e.g. spectral characterization of the fluorescent probe, integration with portable instrumentation, etc. This could make the work more interesting and give it solidity.

The authors should also carefully revise English spelling and text editing, as more than some words are used in an unappropriate meaning.

Additional corrections could be found in the attached pdf file.
